# A Gas Sensors Detection System for Real-Time Monitoring of Changes in Volatile Organic Compounds during Oolong Tea Processing

**DOI:** 10.3390/foods13111721

**Published:** 2024-05-30

**Authors:** Zhang Han, Waqas Ahmad, Yanna Rong, Xuanyu Chen, Songguang Zhao, Jinghao Yu, Pengfei Zheng, Chunchi Huang, Huanhuan Li

**Affiliations:** 1School of Mechanical Engineering, Jiangsu University, Zhenjiang 212013, China; hello1hz@163.com; 2School of Food and Biological Engineering, Jiangsu University, Zhenjiang 212013, China; waqas3ahmad@gmail.com (W.A.); ryn981020@163.com (Y.R.); swchenxuanyu@gmail.com (X.C.); zsgemail9679@163.com (S.Z.); yujinghao878@163.com (J.Y.); 17853260667@163.com (P.Z.); 3Chichun Machinery (Xiamen) Co., Ltd., Xiamen 361100, China; huangchunchi@126.com

**Keywords:** Oolong tea, oxidation, gas sensors, recognition models

## Abstract

The oxidation step in Oolong tea processing significantly influences its final flavor and aroma. In this study, a gas sensors detection system based on 13 metal oxide semiconductors with strong stability and sensitivity to the aroma during the Oolong tea oxidation production is proposed. The gas sensors detection system consists of a gas path, a signal acquisition module, and a signal processing module. The characteristic response signals of the sensor exhibit rapid release of volatile organic compounds (VOCs) such as aldehydes, alcohols, and olefins during oxidative production. Furthermore, principal component analysis (PCA) is used to extract the features of the collected signals. Then, three classical recognition models and two convolutional neural network (CNN) deep learning models were established, including linear discriminant analysis (LDA), k-nearest neighbors (KNN), back-propagation neural network (BP-ANN), LeNet5, and AlexNet. The results indicate that the BP-ANN model achieved optimal recognition performance with a 3–4–1 topology at pc = 3 with accuracy rates for the calibration and prediction of 94.16% and 94.11%, respectively. Therefore, the proposed gas sensors detection system can effectively differentiate between the distinct stages of the Oolong tea oxidation process. This work can improve the stability of Oolong tea products and facilitate the automation of the oxidation process. The detection system is capable of long-term online real-time monitoring of the processing process.

## 1. Introduction

Oolong tea, a distinctive tea category originating from China, has a rich history spanning over three hundred years, dating back to the Yongzheng era of the Qing Dynasty [1]. This tea is renowned for its enduring fragrance [2]. Oolong tea undergoes several crucial processing stages in order to produce its distinctive aroma. Fresh tea leaves, after being harvested, undergo several processing steps including sorting, sun withering, oxidation, fixing, rolling, and roasting. Each step involves specific procedures and standard protocols to ensure the production of high-quality Oolong tea [3]. 

The oxidation step in particular is crucial for the formation of the associated distinctive aroma [4] including alternating and repeating stages of shaking and fermenting in the production of Oolong tea [5]. Friction and collision promote the rupture of leaf margin cells and the diffusion and penetration of water, contributing to the release of a grassy flavor [6]. The subsequent fermentation step involves the settling of tea leaves, with moisture permeating from the stems and veins to the leaf cells [7]. The soluble substances in the tea leaves gradually oxidize and ferment, forming the main chemical components that decisively influence the final product’s quality [8].

Oolong tea has higher levels of ether extracts and phenolic compounds than other teas; this is due to oxidation [9]. Oxidation, as the primary fermentation stage, forms polyphenolic compounds through chemical and enzymatic reactions [8]. During oxidation, the low boiling point species that induce a raw, grassy smell are volatilized and transformed, while the high-boiling-point component that imparts a floral and fruity fragrance persists [10,11]. Therefore, the quality control of the Oolong tea oxidation process is particularly important. Traditional processing heavily relies on the expertise of tea makers, lacking a comprehensive automated control system [12]. Therefore, employing convenient and nondestructive detection technologies to precisely monitor this key step holds promise in tea aroma and flavor quality information [13]. Although the practical application of this research method requires a significant amount of time to build the discrimination model database, it can provide a theoretical basis for further guiding tea production.

To detect the aroma of rooibos tea (*Aspalathus linearis*), Song et al. utilized both simultaneous distillation–extraction (SDE) and reduced-pressure steam distillation (DRP) methods for the extraction of volatile compounds. Subsequently, gas chromatography–mass spectrometry (GC-MS) was employed to analyze the volatile aroma-related compounds in the extracts. Ultimately, 50 volatile compounds were identified in both extraction methods. Comparative analysis revealed that aldehydes and acidic compounds among them contribute to the formation of the unique aroma of tea [14]. Li, ZW et al. employed headspace solid-phase microextraction (HS-SPME) to extract volatile aroma components from five varieties of Fenghuang Dancong tea. They combined gas chromatography–mass spectrometry (GC-MS) and gas chromatography–olfactometry (GC-O) methods for component detection. The detection results revealed that 116 volatile organic compounds (VOCs) were identified through GC-MS, while 26 active VOCs contributing to the aroma of tea were detected based on GC-O analysis [15]. Both GC-MS and GC-O methods exhibited high accuracy and sensitivity. However, the extraction process is time-consuming, and the detection process is cumbersome, which limits its application for rapid detection at tea processing sites. Moreover, the high cost of detection instruments makes it difficult for many tea processing factories to afford them. Therefore, there is a need to develop a detection system for rapid assessment of the tea processing status during the production process.

The gas sensors are rapid, nondestructive, and can effectively differentiate between different types of tea based on aroma profiles. However, they may lack specificity and sensitivity compared to more traditional analytical techniques such as liquid chromatography (LC) and gas chromatography (GC). However, their simplicity and speed make them suitable for applications where rapid or on-site analysis is needed [16]. Gas sensing detection technology, which emerged and developed after the 1990s, utilizes a series of gas-sensitive sensors and recognition algorithms to perceive and identify odors. The basic principle of this detection technology involves using gas sensors with cross-sensitivity to various gases to form a sensor array. This array converts information from mixed gases into electrical signals related to components and concentrations. These signals are then collected by a signal acquisition circuit. After the collected signals have been analyzed and processed by a signal processing module, it becomes possible to classify and identify different gases or determine the proportions of various components in complex mixed gases [17]. Tozlu and Okumuş designed a fermentation system equipped with a gas sensor detection system function to detect the aroma during the fermentation process of black tea to enhance the production capacity of tea gardens [18]. Tseng et al. successfully differentiated samples with significant grassy odor variations during the Oolong tea oxidation process using a gas sensors detection system with metal oxide sensors, demonstrating its feasibility [19]. The response of metal oxide sensors to a given gas can vary across different sensors. Consequently, it becomes particularly important to identify sensors among multiple options that exhibit sensitivity to the specific target gas for precise detection [20]. Wang et al. applied correlation analysis, variance analysis, and cluster analysis for sensor selection [21]. Lu et al. proposed the sensor array optimization (SAO) method using correlation coefficient and cluster analysis (CA). Subsequent experimental validation using a linear discriminant analysis (LDA) method based on averages (LDA-ave) combined with the nearest neighbor classifier (NNC) achieved a classification accuracy of nearly 94.44~100% [22]. These studies provide a basis for the optimization of sensor arrays in gas sensors detection systems. 

In particular, multivariate statistical analysis methods have shown remarkable performance in classification and detection tasks using public datasets [23]. Dutta and others used E-nose technology coupled with machine learning techniques to analyze five different methods of processing black tea, achieving differentiation accuracies of 88% and 89%, respectively [24]. Kang et al. utilized a gas sensors detection system to gather aroma information from tea leaves harvested during different process periods [25]. By integrating the detection system with a classic convolutional neural network for tea leaf identification, they achieved an accuracy of 97.62%. The combination of a gas sensors detection system with multivariate statistical analysis revealed suitability for differentiation gases in tea leaves from different harvesting periods. However, the aroma monitoring of Oolong tea during oxidation has not been reported for a gas sensors detection system combined with multivariate analysis.

Therefore, our study focused on developing a gas sensors detection system combined with multivariate analysis to evaluate aroma quality during the Oolong tea oxidation process. We achieved this according to the following steps: (1) selecting an appropriate sensor array for online monitoring of the Oolong tea oxidation process based on 13 metal oxide semiconductors. (2) examining the relationship between tea aroma data and the oxidation state through linear discriminant analysis (LDA), k-nearest neighbors (KNN), a back-propagation neural network (BP-ANN), LeNet5, and AlexNet. (3) Finally, the proposed model was applied for the identification of the oxidation state to monitor the Oolong tea oxidation process.

## 2. Materials and Methods

### 2.1. Oolong Tea Oxidation Process

This study examines the oxidation process of the tea brand “Iron Goddess of Mercy” from Anxi in southern Fujian. Before designing the experiment, we referred to relevant literature and conducted on-site investigations at tea processing locations [6,7]. Based on documented testing methods and the expertise of tea masters, we determined the processing methods and equipment operating parameters for different stages of the oxidation process of Oolong tea. To systematically and precisely investigate this process, the study monitors two key stages: shaking and fermentation. During the oxidation process of Iron Goddess of Mercy tea, three shaking and three fermentation cycles are implemented. Each shaking and fermentation process is as follows: the first shaking lasts for 5 min at a frequency of 25 Hz and the first fermentation lasts for 30 min; the second shaking lasts for 14 min at a frequency of 25 Hz and the second fermentation lasts for 50 min; the third shaking lasts between 50 to 70 min at a frequency of 20 Hz and the third fermentation lasts for 8–10 h. Given that the first two shaking and fermentation cycles have short intervals and indistinct characteristics, traditional timed processing is applied. Aromas are monitored during the primary processing stages (the third shaking and third fermentation stages). Tea aroma is collected during the third shaking and third fermentation stages, and processing states are classified based on the shaking and fermentation nodes assessed by on-site tea masters during each evaluation. 

### 2.2. Sensor Selection

The signal source of the gas sensors detection system comprises a sensor array, and the main compounds present in the aroma and flavor of Oolong tea during fermentation are alcohols, aldehydes, and hydrocarbons [26]. Therefore, when developing a gas sensors detection system, it is essential to choose sensor types based on the major compounds in the aroma of tea. This strategic selection provides us with the capability to detect nuanced changes in aroma throughout the tea processing stages. In this study, we initially selected 13 metal oxide semiconductor gas sensors with strong stability and sensitivity to VOCs in the aroma of tea (Table 1), based on the aroma components during the fermentation process of Oolong tea [27]. Among them, the TGS series sensors were sourced from Figaro Engineering Inc., Osaka, Japan, while the other sensors were obtained from Zhengzhou Winsen Technology Corp., Zhengzhou, China.

### 2.3. Gas Sensors Detection System Integration

The gas sensors detection system’s core comprises the sensor circuit and reaction chamber, and the stability and reliability of sensor signal readings are determined by the design of the test circuit [28]. The reaction chamber serves as the space where gases interact with the array of gas sensors and undergo reactions. The uniform distribution of gases, gas flow rates, and the state of gas flow in the reaction chamber directly impact the detection results of the sensors. The integration of the sensor test circuit with the rational design of the reaction chamber significantly affects the stability and reliability of the gas sensors detection system’s signal [29].

All the sensors listed in Table 1 are metal oxide sensors, which are divided into four-pin and six-pin sensors based on the structure of the sensor. The circuit diagram of the test circuit is shown in Appendix A. The sensor is internally divided into two circuits: a heating circuit and a working circuit. The design of the working circuit requires calculating the load resistance size for the detection circuit and selecting appropriate circuit components based on the sensor’s technical manual and working principle. The device circuit schematic integrates the sensor, microcontroller, and power circuit onto the same printed circuit board in Appendix A. This highly integrated design optimizes the circuit layout, reducing electromagnetic interference and signal transmission losses, thereby ensuring a stable operating environment and enhancing signal processing efficiency and reliability [30]. 

Simulation of gas transport, diffusion, distribution, and flow patterns within the reaction chamber was conducted using the CFD module of COMSOL Multiphysics 6.0 software [31], as shown in Figure 1. The simulation results aided in the optimization of the reaction chamber’s design [32]. In order to achieve the ideal flow state of gas in the reaction chamber, three draught boards and two septums were added to regulate the gas distribution and flow. This effectively prevented the formation of dead zones (areas of gas stagnation) and circulating regions, contributing to enhanced sensor performance and sensitivity.

### 2.4. System Workflow

The custom-built gas sensors detection system consists of a gas path, a signal acquisition module, and a signal processing module. The internal structure of the device and the schematic diagram of the system detection process are shown in Figure 2. 

The workflow of this device is as follows.

(1)Device initialization: The cleaning air path is connected. The gas sensors detection system undergoes a 40 min preheating process. Afterward, the air path is switched to the sampling path using solenoid valves. The aroma, which is drawn out by the air pump, initially enters the filter, where dust particles and water vapor are removed to ensure a controlled experimental environment.(2)Sampling and detection: The filtered aroma enters the reaction chamber, interacting with the sensor array and initiating a 4 min detection cycle (2 min for the reaction phase and 2 min for the reduction phase). After each detection cycle, the air path is adjusted to the cleaning path to restore the sensor state (Figure 3A) and clear any residual tea aroma.(3)Signal processing: The signals produced by the sensors are rectified and filtered by the sampling circuit, then undergo A/D conversion via the microcontroller’s IO port, converting the detected analog signals into digital signals that can be recognized and processed by software. These digital signals are then transmitted to the PC via USART serial communication for data storage and processing.

**Figure 3 foods-13-01721-f003:**
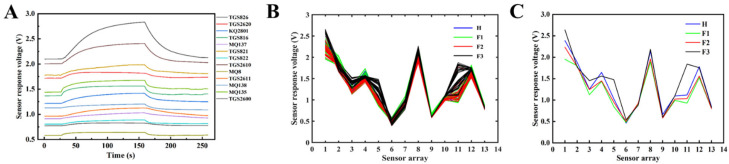
Sensor testing and reduction curve (**A**). Sensor response spectra of 256 samples (**B**) and Four-stages sample line chart (**C**). H1: third shaking phase, F1: de-greening phase, F2: aroma-producing phase, F3: stabilization phase.

A typical sample response curve is depicted in Figure 3A, where each curve illustrates the variation in response values of different sensors over time as the tea aroma permeates the reaction chamber. It can be observed that all sensors exhibit a consistent response pattern within a detection cycle, and the peak response values can be identified. Subsequent experiments indicated that all collected samples in this study share similar characteristics. Therefore, utilizing response values for subsequent feature extraction and pattern recognition within a response cycle is reasonable.

### 2.5. Sample Acquisition 

The patterns of aroma changes during the third fermentation were identified via repeated experiments, including the de-greening phase (transforming and degrading grassy compounds in the tea, reducing grassy odor, and gradually producing the tea aroma), the aroma-producing phase (characterized by a substantial transformation of internal tea solutes, generating the unique aroma of Oolong tea), and the stabilization phase (in which the chemical reactions for aroma substances are nearly complete, and the aroma composition stabilizes, marking the end of the oxidation process). Thus, we divided the experimental stage into four monitoring zones: the third shaking (H1) and three different stages of the third fermentation (F1 de-greening phase, F2 aroma-producing phase, and F3 stabilization phase). The experimental samples were taken from autumn tea leaves, which are known for their high yield and quality, and which were collected between October 15 and 18. The sampling process began at 6 PM each day and ended at 3 AM. Based on the different rates of aroma change and processing duration, the sampling frequency varied for each stage: the H1 stage was sampled every 4 min, the F1 stage was sampled every 8 min, the F2 stage was sampled every 15 min, and the F3 stage was sampled every 10 min. In total, 256 samples were obtained from 4 batches (Figure 3B). The discriminant analysis model was built by integrating experimental data from these different batches. 

### 2.6. Data Analysis Method

The signals captured by sensors frequently exhibit diverse noises, and therefore employing a Butterworth low-pass filter to process the sample data is imperative for eliminating short-term fluctuations or peaks [33]. As the concentration of tea aroma in the samples decreases during the process, this experiment employs a sliding window method to find the peak signal within a sample cycle as the characteristic of the sample, followed by zero-centering the data [34]. Principal component analysis (PCA) is used to ascertain the contribution rates of the main components and their cumulative contributions from the original dataset and extract key information [35]. Before conducting principal component analysis (PCA), we standardized the response values in the data matrix to eliminate the dimensional influence between different sensors. The application of PCA aims to reduce the dimensionality of the data and attempts to reveal the main factors that affect data variability. Through PCA, we identified the principal components with high contribution rates and used these components as inputs for subsequent models to discriminate the oxidation stages of Oolong tea. Specifically, we selected the first two principal components that explained most of the total variance for analysis. Before establishing the discriminant models, the samples were randomly sorted and subsequently divided into calibration and prediction sets in a 3:2 ratio. As a result, 154 samples make up the calibration set, and 102 samples make up the prediction set. Subsequently, five different discriminant methods were systematically employed to formulate identification models. These methods included three classical machine learning methods and two classical CNN deep learning methods: linear discriminant analysis (LDA), k-nearest neighbors (KNN), back propagation neural network (BP-ANN), LeNet5, and AlexNet. These identification models are described in detail in the Appendix A. The cross-validation method was applied to determine the optimal model. The optimal classification model was obtained by verifying the classification performance of the model with the predicted samples. In this study, multivariate analysis was conducted using MATLAB R2014a software (MathWorks Inc., Natick, MA, USA), PyTorch 1.4.0, and Spyder (Python 3.6).

### 2.7. Evaluation Metrics of Results

In evaluating the effectiveness of classification models, performance metrics such as sensitivity, specificity, accuracy, and error rate are commonly used to reflect the actual proportions of classified samples. These metrics are usually extracted from the confusion matrix and serve as an index to examine the classification model efficiency. Sensitivity measures the effectiveness of the model in accurately identifying samples belonging to a specific category. Specificity reflects the model’s ability to correctly exclude non-target category samples. Accuracy and error rate provide a comprehensive ability of the model’s overall proficiency in classifying samples. In the detection process, instances that are correctly identified are labeled as TP (True Positive), while those that are correctly rejected are labeled as TN (True Negative). Instances that are incorrectly identified are labeled as FP (False Positive), and those incorrectly rejected are marked as FN (False Negative). These performance metrics are defined by Equations (1)–(4) below:(1)Sensitivity=TP/(TN+TP)
(2)Specificity=TN/(TN+TP)
(3)Accuracy=(TP+TN)/(TN+TP+FP+FN)
(4)Error rate=1−Acc=(FP+FN)/(TN+TP+FP+FN)

## 3. Results and Discussion

### 3.1. Sample Response Graph Analysis

Figure 3C illustrates sensor response graphs for samples at four different stages. The radar chart in Figure 4 illustrates the aroma detection information of tea leaves at different processing stages, with the central axis representing the sensor’s response values. It can be observed that each sensor responds to the detected samples, indicating the cross-sensitivity of different sensors to the volatile gas. However, there are differences in the response characteristics among different sensors, and their contributions to experimental data vary [36]. Therefore, it is necessary to conduct a comprehensive analysis of sensor response information to enhance the precision of gas information classification.

As indicated by the line graph of samples H and F3, sensors such as TGS826, TGS816, TGS822, TGS2610, TGS2611, MQ135, and TGS2600 undergo changes corresponding to the odor concentration across all four stages. exhibiting characteristic responses to the entire oxidation process’s mixed gases. On the other hand, sensors TGS2620 and MQ8 show a pronounced response to grassy odors, with lower sensitivity to aromas generated throughout the oxidation process. Sensors, KQ2801, MQ137, TGS821, and MQ138 exhibit a significantly improved response to sample F3 compared to sample H. As evident from the response curve line graph and the characteristic response gases of sensors in Table 1, it can be concluded that tea leaves induce the rapid release of aldehydes and alcohols from volatile compounds in green leaves through the shaking process. Fermentation promotes the transformation of soluble substances within green leaves, leading to the accumulation of characteristic alcohols and olefins [37]. Therefore, a preliminary assessment of the tea processing status can be made based on the response curves of characteristic sensors combined with scientific calculation methods.

### 3.2. PCA Results

In the exploration of the oxidation stage of Oolong tea, PCA served as a valuable tool for visualizing clustering potential and extracting pertinent information from the original dataset [38]. The impact of environmental and other variables may introduce noise to the collected data, influencing subsequent model analysis [39]. The application of principal component analysis (PCA) can reduce dimensionality, extract feature information from the samples, and provide a preliminary assessment of inter-class similarities [40]. Specifically focusing on the signals from 13 metal-oxide (MOS) gas sensors, the first ten principal components (PCs) were identified as contributors to 99.96% of the overall variance (Appendix A).

Each of these PCs exhibited eigenvalues surpassing 0.005. This shows their significance in capturing substantial original information from the MOS signals (Figure 5A). The subsequent application of these PCs facilitates monitoring MOS with higher loading values (Figure 5B). The graphical representation of the first two PCs, PC1 and PC2, in Figure 5C revealed a discernible clustering pattern within the oxidation stage of Oolong tea. The first two principal components were selected for feature analysis by calculating their cumulative variance (Table 1). In the subsequent calculations, the first 10 PCs were used as inputs for the model and determined the optimal conditions for model performance through cross-validation. PCA was performed on the signal values obtained from each of these sensors. This analysis allowed us to reduce the dimensionality of data by identifying and selecting PCs that represent the highest variance and contribution rates among the signals [41]. These selected components effectively encapsulate the majority of the information provided by the 13 sensors, ensuring that no critical data were overlooked during our analysis. This approach not only confirms the comprehensive use of all sensors but also enhances the robustness and reliability of the sensor selection methodology.

### 3.3. Recognition Model Results

The outcomes of the LDA analysis revealed identification rates of 83.77% (calibration set) and 83.33% (prediction set) for the oxidation stage of Oolong tea, utilizing six PCs (Appendix A). The majority of identification errors were observed in samples from H and F1 stages (Appendix A). Notably, the specificity of the calibration set (Appendix A) demonstrated a slight improvement, while the sensitivity was comparatively lower.

The identification rate of the LDA model with different PCs is shown in the calibration set and prediction set (A). Sample identification of the optimal parameters of PCs was achieved by cross-validation based on the calibration set (B). Cross-validation results for the prediction set in KNN with varying K values and PCs are depicted in Figure 6A,B. At K = 3 and PCs = 3, the largest identification rate of 88.24% (Appendix A) in the prediction set was attained. The attained values for sensitivity ranged from 0.80 to 1, while the specificity ranged from 0.94 to 0.99 in the prediction set (Appendix A). These values confirmed the enhanced performance of the KNN compared to that of LDA models, particularly for the H, F1, and F2 stages. Specifically, the sensitivity ranged from 0.80 to 1, while the specificity ranged from 0.94 to 0.99 in the prediction set.

This experiment optimized two classic CNN network models, LeNet5 and AlexNet, and compared their network performance. The LeNet5 model underwent training for 500 epochs, as shown in the loss curve (Figure 7A) and model results (Figure 7B). Training reached optimal convergence at 190 epochs, achieving recognition rates of 92.86% and 89.21% for the calibration and prediction sets, respectively (Appendix A). The sensitivity and specificity for the prediction set (Appendix A) were in the ranges of 0.80–0.96 and 0.93–0.99, respectively.

For the AlexNet network model, training was conducted for 200 epochs, as depicted in the loss curve (Figure 7C) and model results (Figure 7D). Optimal convergence was reached at 49 epochs, with recognition rates of 94.16% and 91.17% for the calibration and prediction sets, respectively (Appendix A). The sensitivity and specificity for the prediction set (Appendix A) were in the ranges of 0.92–1 and 0.94–0.99, respectively. Compared to AlexNet, the LeNet5 network has a shallower structure and lower feature extraction capability, resulting in relatively poorer classification results. The AlexNet network model utilized data augmentation and dropout regularization as two methods to prevent overfitting, and thus enhanced its ability to fit the data. In situations with a limited number of samples, it demonstrated superior performance.

In Figure 8, the identification rates of the BPANN model for different PCs in the calibration and prediction sets are presented. The optimal configuration for the BPANN model in identifying the oxidation stage of Oolong tea was determined with PCs set to three. The corresponding identification rates for the calibration and prediction sets were 94.16% and 94.11%, respectively (Appendix A and Figure 8A). The model demonstrated a four-epoch structure in the hidden layer (Figure 8B), resulting in a final BPANN model topology of 3–4–1. Furthermore, six validation checks were conducted during the calibration process (Figure 8C). The calculated sensitivity values for BPANN ranged from 0.92 to 0.96, and the specificity ranged from 0.97 to 0.99 (Appendix A).

This investigation systematically examined a developed MOS sensor and employed five distinct classification models to enhance the identification of the oxidation stage of Oolong tea. Appendix A reveals that the sensitivity and specificity ranges of the prediction set for the BPANN model are 0.92 to 0.96 and 0.97 to 1, respectively, and these are superior to the LAD and KNN models. The AlexNet network model showed higher sensitivity; however, it indicated a lower predictive performance compared to BPANN. 

The comparison of five machine learning models in identifying oxidation stages in Oolong tea shows that the BPANN is the most effective, achieving high recognition rates and balancing sensitivity and specificity well. In the performance evaluation, the order was established as BPANN > AlexNet > LeNet5 > KNN > LDA. As can be seen in Appendix A, the BPANN model with a 94.11% accuracy in predicting the oxidation stage of Oolong tea in the prediction set outperformed the linear model LDA (83.33%), KNN (88.24%), and the deep learning models LeNet5 (89.21%) and AlexNet (91.17%). BPANN outperforms the other models due to its ability to adapt to complex patterns and efficiently utilize PCs to reduce noise and overfitting [42]. AlexNet, while fast, slightly lags behind in accuracy [43]; LeNet5, although accurate, requires longer training times [44]. KNN struggles with lower identification rates and sensitivity to parameter settings, while LDA suffers from high error rates and lower performance metrics, likely due to its assumptions about data distribution [45]. Overall, BPANN’s robustness and accuracy make it particularly suitable for precise tasks like classifying tea oxidation stages.

Therefore, the separation of the oxidation stage of Oolong Tea was complicated, particularly for samples from the H, F1, and F3 stages. It was observed that the components in the oxidation stage of Oolong tea vary and that the trend in change is nonlinear. Models characterized by high nonlinearity, supported by strong self-learning and self-adjustment capabilities, exhibit optimal performance in addressing complex interactions. The results indicate that the BPANN model performs better than other classification models primarily attributed to the inherent topological network structure. This structure may appear to be potentially more favorable for evaluating the oxidation stage of Oolong tea.

## 4. Conclusions

In this study, we developed a biomimetic olfactory gas sensors detection system for monitoring the aroma of Oolong tea leaves during the oxidation process. The gas sensors detection system was combined with multivariate analysis to classify different stages of Oolong tea oxidation. The obtained data were subjected to five classification models, including LDA, KNN, LeNet5, AlexNet, and BPANN. The BPANN model achieved the best recognition results, indicating an identification rate of 94.16% for the calibration set and 94.11% for the prediction set. 

This study provides a scientific and novel approach to enhancing the stability of production quality during the Oolong tea oxidation process and for advancing the automation of this process. However, the current research on guiding the key stages of Oolong tea processing for production is still insufficient, and there is room for improvement in the long-term detection capability of the equipment. Future studies can further enhance the performance of the equipment and utilize the methods proposed to optimize different stages of the third Oolong tea fermentation process. This includes exploring methods to shorten the oxidation time of Oolong tea and increase the yield of characteristic products. 

## Figures and Tables

**Figure 1 foods-13-01721-f001:**
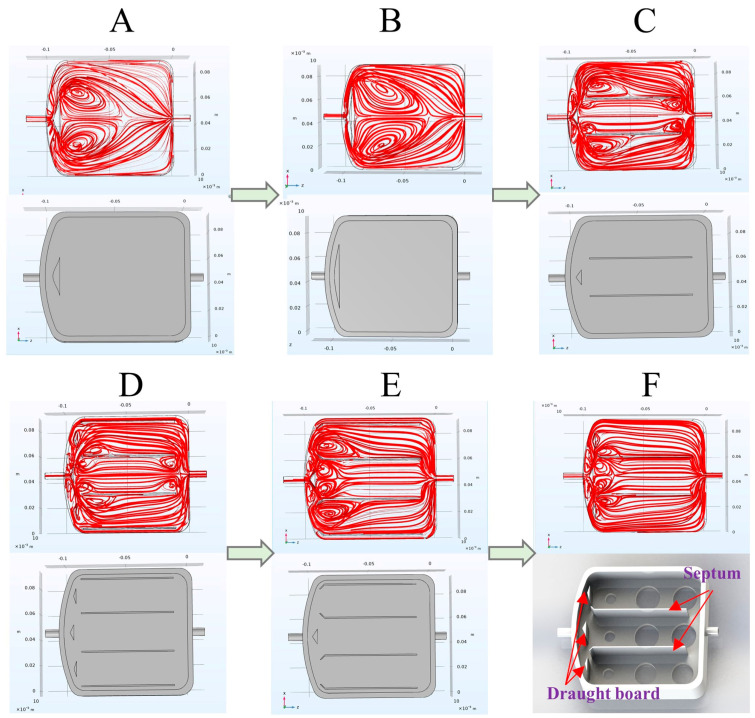
Figures (**A**–**F**) show the improvement process of the structure of the reaction chamber and the velocity flow diagram of each structure. Figure (**F**) shows the optimal model.

**Figure 2 foods-13-01721-f002:**
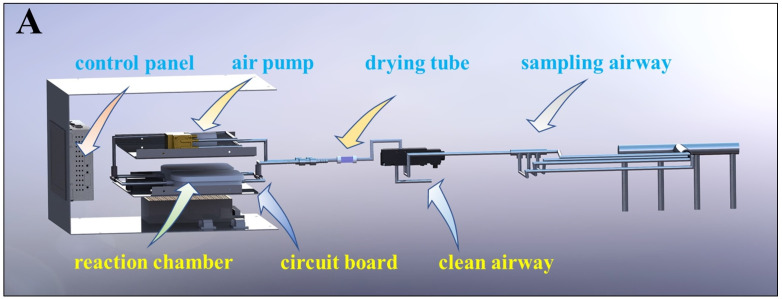
Schematic diagram of internal structure and gas path of gas sensors detection system (**A**). System detection process diagram (**B**).

**Figure 4 foods-13-01721-f004:**
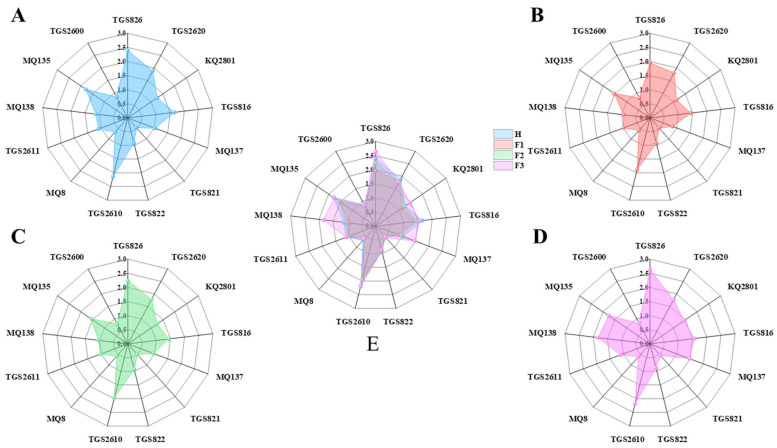
Radar charts for tea aroma detection at different processing stages: (**A**) H, (**B**) F1, (**C**) F2, (**D**) F3, (**E**) overall summary of the four samples.

**Figure 5 foods-13-01721-f005:**
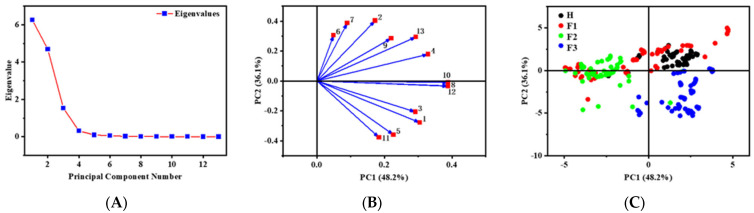
Eigenvalue and of the principal components (**A**). Score plot of sensors (**B**) and loading plot of sensors (**C**). The numbers 1–13 in Figure B represent the 13 sensors listed in Table 1. H1: third shaking phase, F1: de-greening phase, F2: aroma-producing phase, F3: stabilization phase.

**Figure 6 foods-13-01721-f006:**
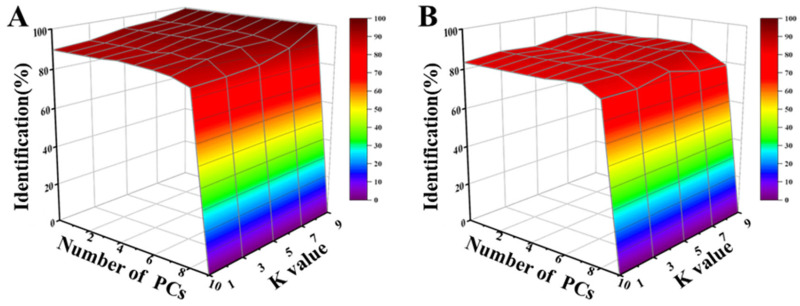
The results of KNN models. Identification rate of KNN model by cross-validation in the calibration (**A**) and prediction (**B**) set according to different PCs and K values.

**Figure 7 foods-13-01721-f007:**
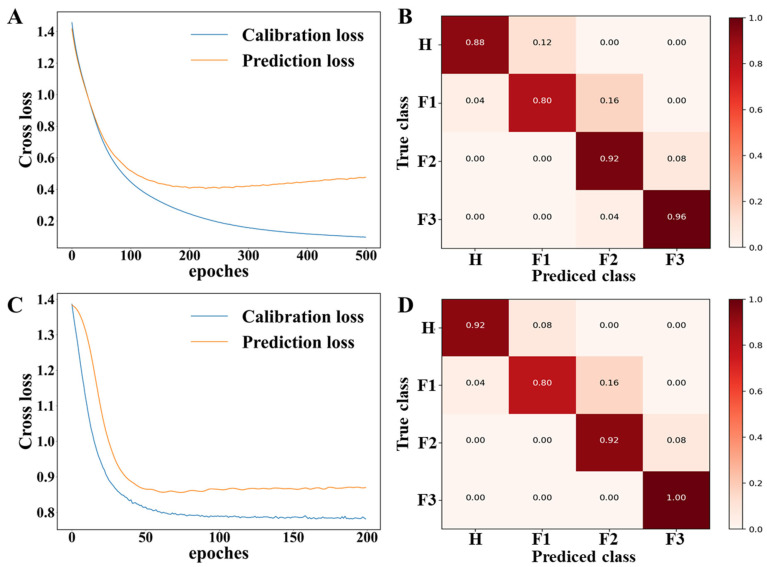
Loss curves and confusion matrices for two CNN deep learning networks: (**A**) Loss curve of LeNet5, (**B**) confusion matrix of LeNet5, (**C**) loss curve of AlexNet, (**D**) confusion matrix of AlexNet.

**Figure 8 foods-13-01721-f008:**
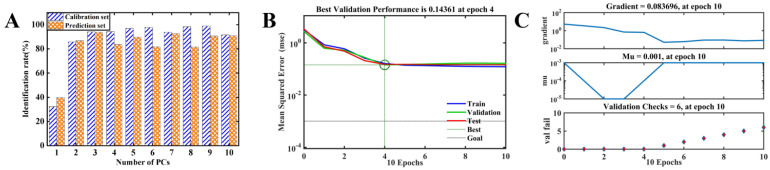
The results of BPANN models. Identification rate of BPANN model with different PCs in the calibration set and prediction set (**A**). Convergence graph of the loss function (**B**). Validation checks for BPANN models (**C**).

**Table 1 foods-13-01721-t001:** Basic information of the selected sensor.

Sensor	Detection Range (ppm)	Sensitivity to Gases
TGS826	30–600	Ammonia gas
TGS2610	200~10,000	LP and gases containing LP components (propane, butane)
TGS821	50–1000	Hydrogen gas
TGS2620	50–5000	Ethanol, organic gas
MQ135	10–1000	Ammonia gas, sulfides, benzene steam
TGS816	50–1000	Methane, propane, butane, alkanes
KQ2801	10–1000	VOC organic gases
MQ138	5–500	Toluene, acetone, ethanol, formaldehyde, organic vapors
TGS2611	300~10,000	Methane, alkane gases
MQ137	5–500	Ammonia gas, organic ammonia gas
TGS822	50–5000	Organic gases, benzyl alcohol, isobutane, acetone
TGS2600	1–500	Hydrogen gas, alcohols
MQ8	100–1000	Hydrogen-containing gas

## Data Availability

The original contributions presented in the study are included in the Article/Appendix A, further inquiries can be directed to the corresponding author.

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
