# Peer review of "A Gas Sensors Detection System for Real-Time Monitoring of Changes in Volatile Organic Compounds during Oolong Tea Processing"

_foods, 2024, doi:10.3390/foods13111721_

Round 1

Reviewer 1 Report

Comments and Suggestions for Authors

This paper offers an interesting perspective on the control of the oxidation process in the production of Oolong tea using an array of gas sensors. The data generated have been analyzed chemometrically, reporting the predictive capacity of LDA, K-nearest neighbors, or neural networks.

General comments

We recognize that the approach is capable of efficiently identifying the elaboration stage but the authors should illustrate the usefulness of this finding. How can these devices control the quality of the product or whether, based on the results obtained, actions can be taken regarding the manufacturing process itself? Comments in this sense would be welcome to improve the overall quality of the paper.

I agree with the authors that the data from this type of sensor is prone to suffer from non-linearity, which can be compensated with more factors or more complex neural structures. However, I am concerned about the ability of neural networks to overfit because I guess that the predictions are not from an independent subset (but are cross-validation?). In this sense, it is unclear how many samples, batches, replicates, etc. have been analyzed (and if two truly independent data sets are available). These issues must be detailed in the experimental part.

Why has one of the most popular classification methods, such as PLS-DA, not been compared? I suggest that, if possible, its effectiveness be tested.

A comparative summary table or figure of the performance of the different modeling methods would be interesting for the results discussion section.

There are some formatting and typo errors throughout the manuscript that can be easily corrected.

Specific comments

Define CNN in the abstract.

Introduction: a more general view can be given about other tea characterization approaches based on other analytical techniques (for example, liquid or gas chromatography) and comment on the pros and cons regarding the version based on gas sensors.

Figure 1 may not be appropriate for Foods readers. It is difficult to interpret the scheme of the integrated circuit. I suggest moving on to supplemental material.

The legends of Figure 4 have not been defined. It should be noted that in A, the acronyms refer to the sensors listed in Table 1. In B and C, I understand that F1, F2,... refer to the process steps indicated in 2.5, but they should appear in the figure caption.

The phrase on line 265 doesn't make much sense. Figure 6a demonstrates that no more than 3 or 4 PCs are necessary to describe all the variance.

Figures 6b and 6c confuse me a bit. I think B refers to the 13 gas sensors and, therefore, would be loading. On the other hand, C would be the different samples, although legends 1, 2, 3, and 4 should be homogenized with the previous ones H, F1, F2, and F3. Given the number of points represented, it is not clear to us as to the number of samples analyzed (I would understand that 4 points, one of each color, would correspond to a processed batch). Are they different, replicated batches??? This point must be clarified.

Fig 7. The quality of the text of the axes must be improved.

Figure 7C needs to be better explained and/or redesigned. If it is a classification graph in which the samples are assigned to the 4 types, how can we know which ones have been classified correctly or incorrectly? In any case, it can also be deleted

In the confusion matrices in Figure 9, I would understand that the 4 categories were H, F1, F2, and F3, but the symbols 0%, 3%, 5%... appear instead. Change or justify correspondingly.

Comments on the Quality of English Language

There are some formatting and typo errors throughout the manuscript that can be easily corrected.

Reviewer 2 Report

Comments and Suggestions for Authors

Dear authors,

The manuscript entitled "A gas sensors detection system for real-time monitoring of VOC change during Oolong tea processing" evaluated the rapid release of volatile compounds aldehydes, alcohols and olefins during the oxidative production of tea by a gas sensors detection with principal component analysis (PCA). It presents scientific relevance for Health, Chemistry, Medicine, Pharmacy and others area. The language (English) is satisfactory (but, I suggest the final revision)! However, you need to change some details/information in the Title, Abstract, Introduction, Material and Methods, results and discussion, and conclusions.

1. Title: Please, to insert the meaning of the acronym VOC? Would they be volatile organic compounds?

2. Abstract: Adequate! But:

- The abstract is well written, but I suggest inserting the methods parameters and results obtained (numerical data!!!) more relevant. Please, to insert the meaning of the acronym CNN.

- At the end, I suggest highlighting the advantages/ disadvantages of the system and methods.

- Keywords: The word "Pattern recognition" does not appear in the title or abstract. Would it really be significant? Was the purpose of the study to "Pattern recognition"? I suggest review!

3. Introduction section:

- I suggest expanding the text a little further, including information about oxidation process”; “gas sensors detection system with metal oxide sensors”; and “analytical techniques”.

- Also, to highlight the "innovative" proposal of the methods, as well as the advantages/disadvantages/limitations of the study.

4. Materials and methods section: The methodological proposal is appropriate to the manuscript, but I suggest:

- Page 2, in “2.1. Oolong Tea Oxidation Process” section: Why does this brand specify (tea brand “Iron Goddess of Mercy”)? Are there no other brands available? I suggest entering more information about the sample used! What are the conditions for collection/acquisition and storage of samples? What is the time/period (from acquisition to analysis)? This information is important to understand the results regarding the oxidation process!

- Page 2, in “2.1. Oolong Tea Oxidation Process” section, lines 96-101: Please enter the reference/protocol used!

- Page 3, in “2.2. Sensor selection”: Were all 13 metal oxide semiconductor gas sensors used? I suggest informing in the text!

- Were figures 1, 2 and 3 produced by the authors? If not, provide references! Page 6, line 157, in “2.4. System workflow” section: Was the workflow produced by the authors? If not, provide references!

- Page 7, in “”section: In abstract (lines 20-21) the authors wrote “…has superior recognition performance with an accuracy of 94.16% in the calibration set and 94.11% in the prediction set…”. But, there is no information about the figures of merit in the " Materials and Methods" section. I suggest relocating the "3.2. Evaluation metrics of results" section (Page 8) to the methods!

- Page 7, in “2.6. Data analysis method” section: I suggest entering more information on how the Principal component analysis (PCA) was applied! How was the data matrix constructed? Could Hierarchical Cluster Analysis (HCA) be used?

5. Results and Discussion

I suggest expanding the discussions - better describe the findings and compare them with other works published in the literature! In particular, for “3.3. PCA result” and “3.5. Results of two CNN deep learning network models”.

- Page 11, in “3.6. Discussion of the results” section: I suggest removing this subtitle! Discussions must be aligned with the previous subtitles. The authors chose to write the topic "3. Results and discussions" (line 219) together. Therefore, for each subsection, the results must be discussed and compared with the literature!

- I suggest, at the end of the "results and discussion", to write a paragraph summarizing the findings and their impacts on the research proposal.

6. Conclusion: I suggest pointing out the main results and disadvantages/limitations of the method and the study!

7. Table and Figures: Adequate.

8. References: Please, check if the references are in accordance with the journal's rules.

Comments on the Quality of English Language

The language (English) is satisfactory (but, I suggest the final revision)! 

Round 2

Reviewer 2 Report

Comments and Suggestions for Authors

The authors improved the manuscript based on my comments!

Comments on the Quality of English Language

Adequate! I only suggest one final review!
